# Selenium catalysis enables negative feedback organic oscillators

Xiuxiu Li[1,2], Polina Fomitskaya[1], Viktoryia A. Smaliak[1], Barbara S. Smith[3], Ekaterina V. Skorb[4] & Sergey N. Semenov [1] ✉

The construction of materials regulated by chemical reaction networks requires regulatory motifs that can be stacked together into systems with desired properties. Multiple autocatalytic reactions producing thiols are known. However, negative feedback loop motifs are unavailable for thiol chemistry. Here, we develop a negative feedback loop based on the seleno-carbonates. In this system, thiols induce the release of aromatic selenols that catalyze the oxidation of thiols by organic peroxides. This negative feedback loop has two important features. First, catalytic oxidation of thiols follows Michaelis-Menten-like kinetics, thus increasing nonlinearity for the negative feedback. Second, the strength of the negative feedback can be tuned by varying substituents in selenocarbonates. When combined with the auto-catalytic production of thiols in a flow reactor, this negative feedback loop induces sustained oscillations. The availability of this negative feedback motif enables the future construction of oscillatory, homeostatic, adaptive, and other regulatory circuits in life-inspired systems and materials.

Living matter is highly adaptable to its environment because of the complex biochemical circuits that regulate it[1,2]. Constructing synthetic materials with comparable levels of adaptivity to their environment requires developing synthetic chemical regulatory circuits[3–10]. However, this is a challenging task, given the complexity and nonlinearity of regulatory reaction networks. Nevertheless, these chemical regulatory circuits are composed of a limited number of individual functional motifs, providing a reasonable ground for modular design[11–14]. These motifs involve oscillators and homeostatic motifs[12], which enable the organism to make psychophysiological adjustments in response to external variations.

Oscillations control many important aspects of cell physiology[15], including circadian rhythms[16], DNA synthesis, mitosis[17,18], and the development of somites in vertebrate embryos[19]. Homeostasis ensures the stability of essential biological parameters such as body temperature, blood sugar levels, and blood pressure. A key feature of both oscillations and homeostasis is negative feedback[11,12]. In principle, any

reaction that removes an autocatalyst can be negative feedback. However, the negative feedback formed by a trivial first-order reaction with an autocatalyst has limited utility for constructing dynamic reaction networks[20]. Such systems are often trapped in a steady state due to the lack of delay or nonlinearity.

For example, sustained oscillations cannot be achieved by combining quadratic autocatalysis with trivial negative feedback[21]. In contrast, combining autocatalysis with a negative feedback loop – that is the situation where an autocatalyst triggers its own elimination – results in robust oscillations, as seen in predator-prey networks[22,23]. Negative feedback loops occur naturally in biochemistry due to the sequence of the orthogonal enzymatic reactions[24,25]. However, designing negative feedback loops in synthetic organic chemistry, where the reactions are not orthogonal and often simply follow the mass action law, is challenging. A notable example involves a hydrazone-based negative feedback loop responsive to zinc ions[26].

[1]Department of Molecular Chemistry and Materials Science, Weizmann Institute of Science, Rehovot, Israel. [2]Department of Chemistry and Shenzhen Key Laboratory of Small Molecule Drug Discovery and Synthesis, Southern University of Science and Technology, Shenzhen, China. [3]School of Biological and Health Systems Engineering, Arizona State University, Tempe, AZ, USA. [4]Infochemistry Scientific Center, ITMO University, Saint Petersburg, Russia. ✉e-mail: sergey.semenov@weizmann.ac.il

**a**  Published oscillators based on small organic molecules

**b**  This work

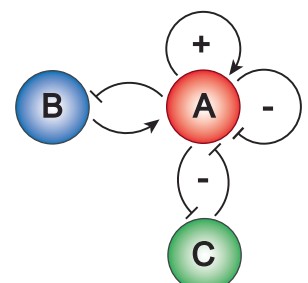
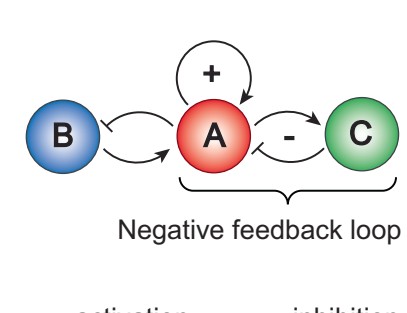

Negative feedback loop

→ activation        ⊣ inhibition

**Fig. 1 | Network topologies.** The difference in network topologies of all previously published oscillators based on small organic molecules (**a**) and oscillators presented in this work (**b**). Red, blue, and green colors represent autocatalysts, substrates, and inhibitors correspondingly. Sharp arrows represent activation; dull arrows represent inhibition.

We have recently developed a series of thiol-based oscillators[27–29], which are called Semenov-Whitesides oscillators by Epstein, Gao, and coworkers[30]. All of them have the same topology, which includes quadratic autocatalysis combined with negative feedback based on two inhibitors (Fig. 1a). Both inhibitors (Michael acceptors or oxidants) directly react with autocatalysts (thiols) according to the mass-action law and do not form a negative feedback loop where autocatalysts should first activate inhibitors from dormant precursors. These direct reactions are neither delayed nor nonlinear; oscillations are achieved by having two inhibitors with drastically different reactivities: one reacting with thiols very quickly, whereas the other reacting slowly[30]. Many non-biological chemical oscillators, including large classes of redox and pH oscillators[31–35] and a recently published amine-based oscillator[36], rely on the direct removal of autocatalysts by inhibitors without a negative feedback loop. Nevertheless, to expand the variety of dynamic behaviors and responses achievable with thiol-based reaction networks, it is essential to enrich the toolbox of available network motifs with negative feedback loops where thiols would trigger their elimination (Fig. 1b)[12]. These motifs would provide negative feedback that is more nonlinear than direct inhibition and enable oscillators with tunable frequency[11,18], adaptive systems[12], and various types of chemical waves and patterns that are instrumental for the synthesis of life-inspired materials[37,38].

Motivated by this question, we explored the selenium-catalyzed oxidation of thiols with organic peroxides. This reaction enabled the construction of negative feedback loops that, when combined with autocatalysis (Fig. 1b), can induce sustained oscillations.

## Results

### Designing negative feedback

To construct a tunable negative feedback loop for thiol chemistry, we must design a reaction network in which thiols would induce their own destruction. This behavior can be achieved when thiols induce the release of a compound that irreversibly reacts with them or of a catalyst that catalyzes an irreversible reaction with them. The latter route is preferable because the amount of the released catalyst can be a small fraction of the amount of thiols in the system. Therefore, if thiols are consumed by the reaction that releases catalysts, this consumption will have a minimal effect on the system.

Thiols are both good nucleophiles and reductants. However, redox reactions are typically more susceptible to catalysis than are nucleophilic reactions and thus are more desirable for negative feedback. Thiols are oxidized by a large variety of compounds; however, most of the oxidation reactions are too fast to be compatible with the minute time scale of the autocatalytic step. Some suitably slow oxidants are bulky organic hydroperoxides (e.g., *tert*-butyl hydroperoxide) and some organic dyes (e.g., methylene blue)[39]. The

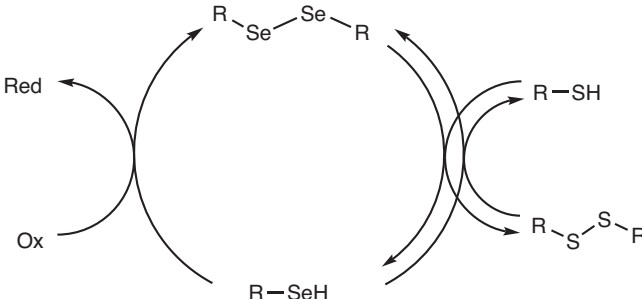

**Fig. 2 | Generic mechanism for the oxidation of thiols catalyzed by selenols.** The scheme neglects mixed seleno-sulfide (RSSeR) and other intermediates of selenide oxidation.

hydroperoxides are preferable because of their high solubility and optical transparency. Therefore, we decided to use the oxidation of thiols by *tert*-butyl hydroperoxide (**1**) as the thiol-removing reaction.

Selenium derivatives are among the most effective catalysts for the oxidation of thiols[39–42]. Rhead and Schrauzer showed that selenite ($SeO_3^{2-}$) can accelerate the oxidation of thiols by methylene blue by two orders of magnitude[39]. Selenite is not an active catalyst, but it is reduced in situ to molecular selenium and other compounds containing Se–Se bonds, which are the active catalysts. Actually, most diselenides are excellent catalysts for this reaction, presumably through the generic mechanism depicted in Fig. 2[39]. The essence of the mechanism is that selenols are oxidized much faster than thiols, and that a very fast thiol-diselenide exchange can quickly regenerate selenols[43].

We chose 4,4′-diselanediyldibenzoic acid (**2**) (Fig. 3a) as the basic catalyst for our study because of three reasons: (i) the convenience of synthesizing aromatic selenols compared to aliphatic selenols, (ii) the solubility in aqueous buffers, and (iii) the expected higher oxidation potential of aromatic diselenides compared with aliphatic diselenides. The molecule was synthesized from *tert*-butyl 4-aminobenzoate by first converting it to *tert*-butyl 4-selenocyanatobenzoate[44], then to di-*tert*-butyl 4,4′-diselanediyldibenzoate, and finally to **2** (see Section 2 of the Supplementary Information for the synthetic procedures).

The oxidation of thiols by **1** that is catalyzed by **2** provides a negative feedback from thiol's production because of the positive dependence of the oxidation rate from the thiol concentration; however, it is not a negative feedback loop because inhibitor species **1** and **2** are not produced by thiols. To create a negative feedback loop, we synthesized a series of selenocarbonates derived from 4-carboxyselenophenol (**7**) (Fig. 3a). These molecules release **7**, an active catalyst, in the exchange reaction with thiols (Fig. 3b). We noted

that the rate of the exchange reaction with cysteamine (**6**) for this class of molecules is in a minute time scale, which is an excellent match with the time scale of the autocatalytic step. To tune the responsiveness of the negative feedback, we varied the substituent on the molecules from methyl (**3**), to ethyl (**4**), and *iso*-propyl (**5**). Expectedly, the rate of the reaction gradually dropped from **3** to **5**; therefore, we can expect a proportional drop in the strength of the negative feedback when changing molecules from **3** to **4** and to **5** (Fig. 3b).

## Constructing oscillators

Sustained chemical oscillations are out-of-equilibrium phenomena. To achieve them in experiments, we must constantly supply reactants and remove products[20]. In this work, we used a micro continuously stirred tank reactor (micro-CSTR) previously described by our group[27]. It has three inlets, one outlet, and a magnetic stirrer (Fig. 4a). The thiol content in the outgoing solution was determined by derivatization with Ellman's reagent followed by analysis in the UV–Vis flow cell (Supplementary Fig. 4). The reactants supplied to the CSTR were thiocholine-based thiouronium salt (**8**), cystamine (**9**), **1**, and one of the catalysts **2–5**. The reaction was carried out in 1 M Tris buffer pH 7.7 at 25 °C.

The full oscillatory reaction network consists of the autocatalytic loop coupled to the negative feedback. Negative feedback can be either by direct oxidation of thiols catalyzed by **2** or a negative

**Fig. 3 | Selenium derivatives used in this study and their reactions with thiols.**
**a** Derivatives of 4-carboxyselenophenol that were used as catalysts in this study.
**b** Reaction of selenocarbonates **3–5** with **6** and its kinetics. Second-order rate constants are calculated based on three independent $^1$H NMR kinetic measurements (Supplementary Figs. 5–7). Errors represent standard deviations based on three independent experiments.

**Fig. 4 | Selenium-catalyzed oxidation-based oscillators. a** A schematic representation of the CSTR experimental set-up and the downstream derivatization with Ellman's reagent. The UV–Vis spectrometer detects the absorption of 4-nitro-3-carboxythiophenolate at 412 nm. **b** The reaction networks of selenium-catalyzed oxidation-based oscillators. The scheme describes oscillators with negative feedback consisting of the thiol-induced release of **7** from **3**, **4**, or **5**. The oscillator with catalyst **2** has **2** in place of **7** and does not have a closed negative feedback loop. Colors represent key elements of the oscillatory network: blue−substrates, red−autocatalysts, and green – inhibitor. Bold plus and minus signs represent positive and negative feedback loops correspondingly.

feedback loop where the catalyst **7** is released from precursors **3**–**5** (Fig. 4b). We used concentrations of **8** (56 or 64 mM) and **9** (100 or 114 mM) that are similar to the previously published oscillator[27]. **1** (73 or 83 mM) was in sufficient excess to avoid the strong effect of its depletion during oscillations, considering the 1:2 stoichiometry of its reaction with thiols. The main adjustable parameters were the flow rate and the concentrations of catalysts **2**–**5**.

Since the behavior of the reactions depends on both the volume of the reactor and the flow rate, we used the ratio of the flow rate to the reaction volume (f/v), also known as the space velocity, to describe the experimental conditions[29]. Interestingly, the oscillators presented here required almost ten times higher flow rates ($f/v = 1.5$–$1.7 \cdot 10^{-3}$ s$^{-1}$) to achieve sustained oscillations than did Semenov-Whitesides oscillators that used two inhibitors ($f/v \sim 2 \cdot 10^{-4}$ s$^{-1}$)[27,29]. This change is not incidental; the fast inhibitor in a two-inhibitor system suppresses autocatalysis, and it is supplied with incoming flow. Thus, the fast flow rates completely suppressed autocatalysis in these systems[27-29]. In the present system, the fast inhibitor is absent, and oscillations occur at much higher flow rates.

The concentrations of **2**–**5**, which are required to obtain sustained oscillations (Fig. 5), follow the expected trend. **2** is used in the lowest concentration because it is an active catalyst as supplied. The required

concentrations for **3**–**5** increase from 0.5 to 2 mM in accordance with the rates of their reactions with thiols (Fig. 3b). The oscillation periods range from 22 to 32 minutes, which is about six times shorter than those of oscillators with two inhibitors[27-29]. This difference is mostly related to the higher space velocity in the present oscillator than in oscillators with two inhibitors.

## Explaining sustained oscillations

The oscillations in the system with only catalyst **2** are surprising. A single inhibitor that removes autocatalysts according to mass action kinetics is insufficient to induce oscillations when combined with quadratic autocatalysis[21]. Moreover, even a negative feedback loop without an autocatalyst-independent delay step is often unable to induce oscillations according to Huck and coworkers[24].

To understand the emergence of oscillations with the catalyst **2**, we studied the kinetics of the oxidation of thiol **6** by hydroperoxide **1**. First, we investigated the kinetics of this reaction without catalysis (Supplementary Fig. 10). The reaction is first-order in **6** and **1**, with a total second order. This rate equation corroborates well with hypotheses that heterolytic splitting of O-O bond by thiolate is the rate-limiting step of this reaction.

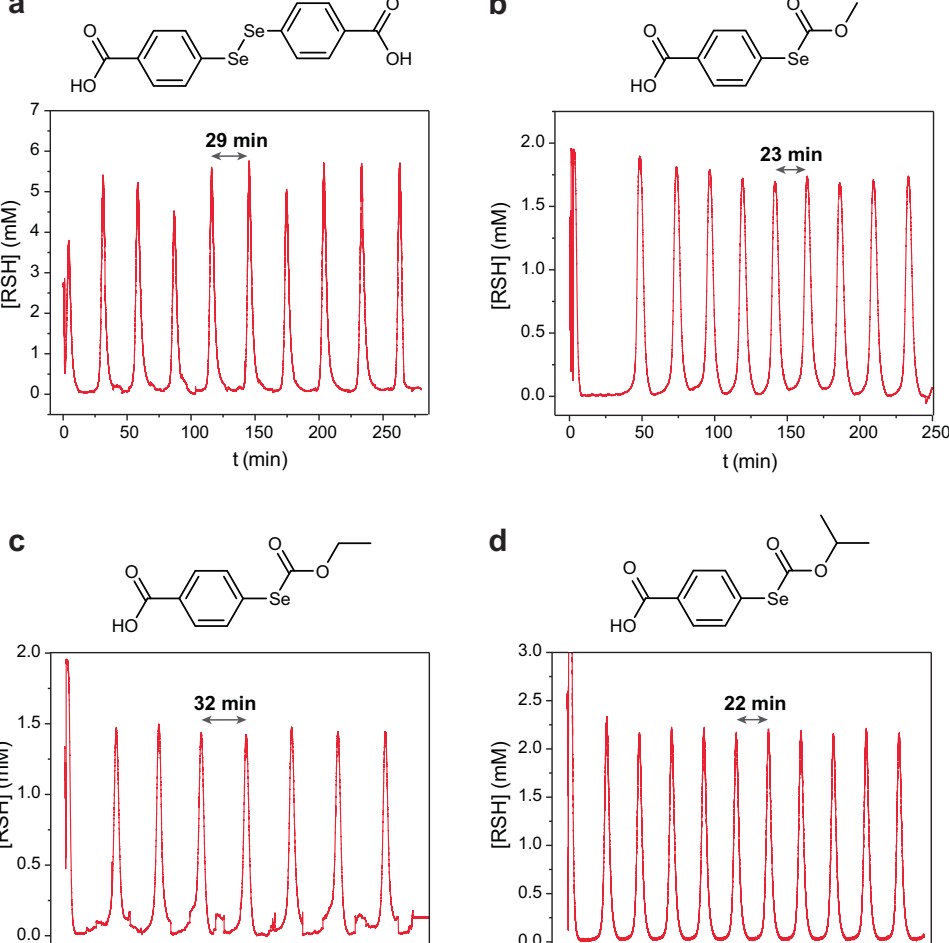

**Fig. 5 | Kinetic studies of the oscillators.** Experimental conditions shared by all experiments: H$_2$O, 1 M Tris-buffer pH 7.7, at 25 °C; other parameters are indicated for specific experiments. **a.** Experimental data showing sustained oscillations in the system with catalyst **2**. [**8**] = 64 mM; [**9**] = 114 mM; [**1**] = 83.2 mM; [**2**] = 0.14 mM; $f/V = 1.46 \cdot 10^{-3}$ s$^{-1}$. **b.** Experimental data showing sustained oscillations in the system with catalyst **3**. [**8**] = 56 mM; [**9**] = 100 mM; [**1**] = 72.8 mM; [**3**] = 0.5 mM; $f/V = 1.67 \cdot 10^{-3}$ s$^{-1}$ **c.** Experimental data showing sustained oscillations in the system with catalyst **4**. [**8**] = 56 mM; [**9**] = 100 mM; [**1**] = 72.8 mM; [**4**] = 1 mM; $f/V = 1.67 \cdot 10^{-3}$ s$^{-1}$ **d.** Experimental data showing sustained oscillations in the system with catalyst **5**. [**8**] = 56 mM; [**9**] = 100 mM; [**1**] = 72.8 mM; [**5**] = 2 mM; $f/V = 1.67 \cdot 10^{-3}$ s$^{-1}$. RSH stands for the sum of the concentrations of all three thiols in the system. Chemical structures of catalysts **2**-**4** are shown above corresponding plots.

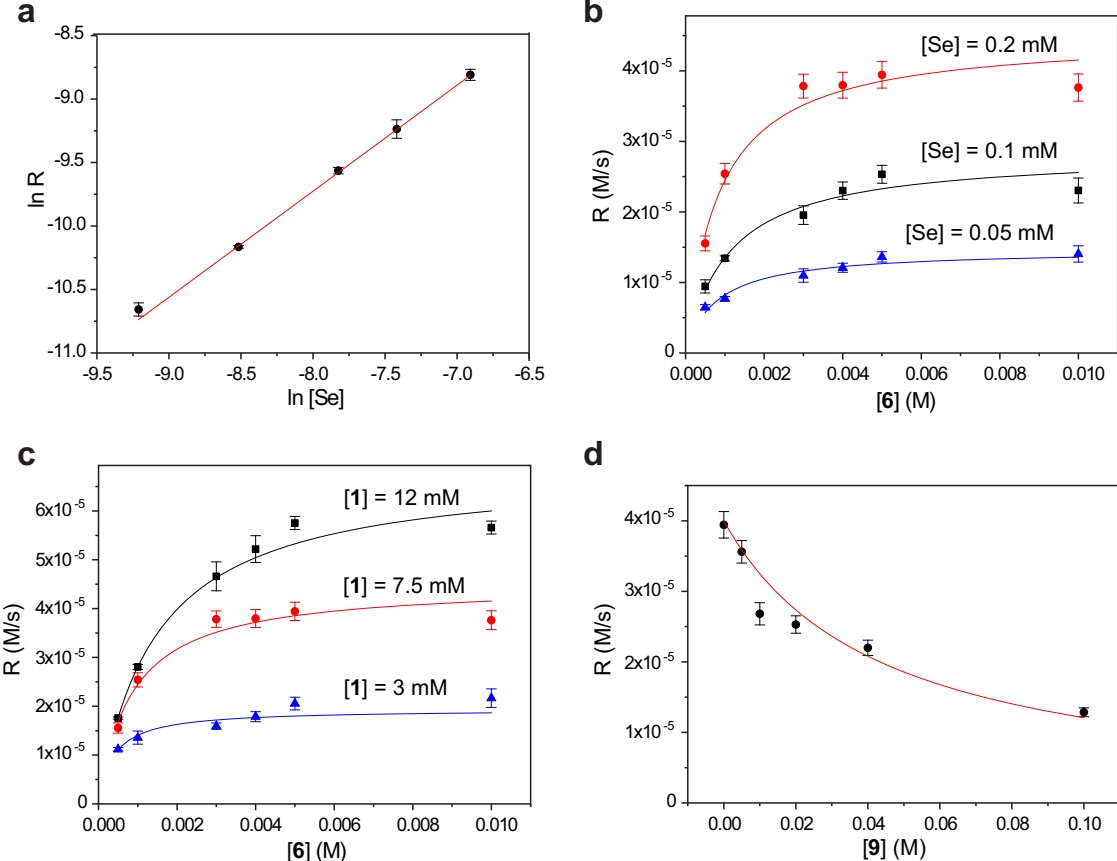

**Fig. 6 | Kinetic studies of the oxidation of 6 by 1 catalyzed by 2. [Se] = 2·[2].** All experiments were conducted in 1 M Tris pH 7.5, at 25 °C. **a** Dependence of the reaction rate from [Se]. **[1]** = 7.5 mM, **[6]** = 5 mM, [Se] = 0.1–1 mM. The dependence in logarithmic coordinates was fit linearly by equation $y = a + bx$; a = −3.0 ± 0.2, b = 0.84 ± 0.02. **b** Dependence of the reaction rate on the concentration of **6** in experiments with different [Se] concentrations. **[1]** = 7.5 mM, **[6]** = 0.5–10 mM, [Se] are shown above each graph. Data for each set were fitted by equation $r = V_{max}·S/(K_M + S)$: [Se] = 0.05 mM, $V_{max} = 1.5 ± 0.1·10^{-5}$ M s$^{-1}$, $K_M = 7.8 ± 1.6·10^{-4}$ M; [Se] = 0.1 mM, $V_{max} = 2.8 ± 0.2·10^{-5}$ M s$^{-1}$, $K_M = 11 ± 1.6·10^{-4}$ M; [Se] = 0.2 mM, $V_{max} = 4.5 ± 0.2·10^{-5}$ M s$^{-1}$, $K_M = 8.4 ± 1.5·10^{-4}$ M. (See Supplementary Fig. 11 for the full range of [Se].)

**c** Dependence of the reaction rate on the concentration of **6** in experiments with different concentrations of **1**. [Se] = 0.2 mM, **[6]** = 0.5–10 mM, **[1]** is shown above each graph. Data for each set were fitted by equation $r = V_{max}·S/(K_M + S)$: **[1]** = 3 mM, $V_{max} = 1.9 ± 0.1·10^{-5}$ M s$^{-1}$, $K_M = 3.8 ± 0.8·10^{4}$ M; **[1]** = 7.5 mM, $V_{max} = 4.5 ± 0.2·10^{-5}$ M s$^{-1}$, $K_M = 8.4 ± 1.5·10^{-4}$ M; **[1]** = 12 mM, $V_{max} = 6.9 ± 0.3·10^{-5}$ M s$^{-1}$, $K_M = 14.5 ± 1.0·10^{-4}$ M. **d** Inhibition of the reaction by **[9]**. **[1]** = 7.5 mM, **[6]** = 5 mM, [Se] = 0.2 mM, **[9]** = 0–100 mM. Data were fitted by equation $y = 2.2·10^{-7}/(0.006 + ax)$, a = 0.11 ± 0.01. Error bars in all plots represent standard deviations based on three independent experiments.

Next, we studied the same reaction catalyzed by **2** (Fig. 6). Expectedly, the rate is proportional to the concentration of **2** (Fig. 6a). The logarithmic plot indicates the reaction order of 0.84 in the catalyst (for generality, we presented data in terms of the selenium concentration [Se] = 2 · **[2]** rather than in terms of concentration of diselenide **2**). We would expect the first order in [Se]; the deviation might result from the complexity of the reaction mechanism or from an experimental error.

Next, we investigated the dependence of the reaction rate on the concentration of **6** at different concentrations of **2** (Fig. 6b). Interestingly, the dependence has a characteristic Michaelis–Menten shape with the reaction rate reaching a plateau at a high concentration of **6**. Fitting these dependencies by the Michaelis–Menten equation $r = V_{max} · S/(K_M + S)$, where $r$ is the reaction rate, $V_{max}$ is the maximum reaction rate, $S$ is the concentration of a substrate, and $K_M$ is the Michaelis constant shows that $V_{max}$ is proportional to [Se], whereas $K_M$ does not significantly depend on it.

To rationalize this behavior, we developed a hypothesis for the reaction mechanism (Fig. 7). It involves the two-stage reversible oxidation of thiols to disulfides by diselenides, followed by the regeneration of the diselenides through the oxidation of selenols with **1**. This mechanism involves two strong simplifications. First, we assumed that

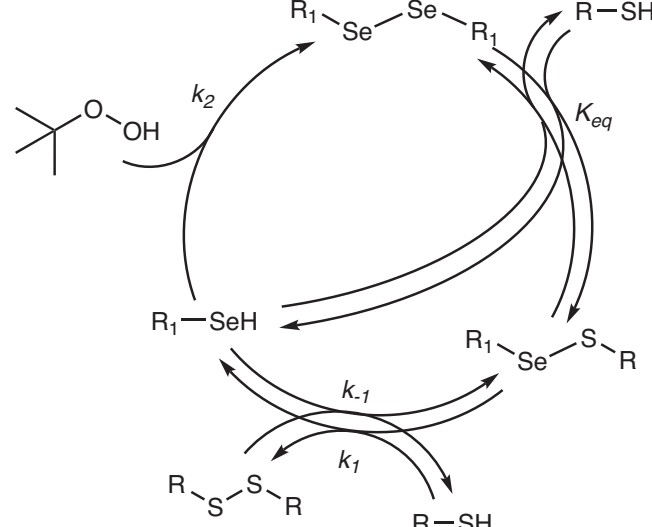

**Fig. 7 | Mechanistic proposal.** Proposed mechanism for the oxidation of thiols with *tert*-butyl hydroperoxide catalyzed by selenophenoles.

selenide is oxidized by **1** directly to diselenide, neglecting the selenenic intermediate (RSeOH)[42] and oxidation to mixed seleno-sulfides (RSSeR) in this step. Second, we assumed the thiol-diselenide exchange to be at equilibrium throughout the reaction. This approximation was partially validated by NMR studies that indicated that this exchange is fast at the NMR timescale (see Section 5 of the Supplementary Information).

The analysis of this mechanism within steady-state approximation results in Eq. (1) (see Section 5 of the Supplementary Information for the derivation):

$$\frac{d[RSH]}{dt} = -\frac{k_2^* Se_0[RSH][tBuOOH]}{k_{-1}/k_1[RSSR] + k_2^*/k_1[tBuOOH] + [RSH]}, \text{where } k_2^* = 2k_2$$

(1)

This equation has the Michaelis–Menten form with respect to the concentration of thiols and explains the dependence of the rate from [Se]. It immediately leads to two additional important consequences: (i) the apparent Michaelis constant should depend on the concentration of **1**, (ii) the reaction should experience product inhibition. We experimentally tested both consequences.

We studied the dependence of the reaction rate from the concentration of **6** in experiments with different concentrations of **1** (Fig. 6c). The fitting of these dependencies by the Michaelis–Menten equation showed that the apparent Michaelis constant $K_M$ increases with an increase in the concentration of **1** in consistency with the equation. We also tested the hypothesis about product inhibition (Fig. 6d). These experiments showed that the rate of oxidation is significantly suppressed by a high (>10 mM) concentration of **9**.

Overall, the fitting of the presented experiments provided five data points with $V_{max}$ and $K_M$ values (Fig. 6a–c). Combining these data with Eq. (1) allowed us to estimate $k_2^* = 33 \pm 5\,s^{-1}M^{-1}$ and $k_2/k_1 = 0.12 \pm 0.02$. The fitting of the substrate inhibition curve gave an estimate for $k_{-1}/k_1 = 0.11 \pm 0.01$ (Fig. 6d).

Inhibitor removal that follows Michaelis–Menten kinetics could enable sustained oscillations[21]. To test this prediction specifically for our oscillating reaction network, we constructed a numerical model of the oscillator COPASI software[45] (see Section 6 of Supplementary Information) and the estimated rate constants for the reactions involved in the autocatalytic loop and for the noncatalytic oxidation of **6** by **1** (Supplementary Fig. 10). Using Eq. (1) and rate constants close to the experimental values obtained for the oxidation of **6**, we were unable to generate sustained oscillations. Nevertheless, the reaction network described here involves three thiols: **6**, thiocholine, and 2-mercaptoethylguanidine (Fig. 4). The kinetics of thiocholine oxidation is particularly important for stabilizing oscillations, because thiocholine, which is released by the hydrolysis and aminolysis of **8**, triggers autocatalysis. Its fast interception might suppress autocatalysis long enough to enable a refill of the reactants between spikes[29].

To determine whether thiocholine oxidation kinetics differs from oxidation kinetics of **6**, we studied the oxidation of thiocholine with **1** in the absence and presence of catalyst **2** in the same conditions as we did for **6** in Tris buffer, pH 7.5, 25 °C. Despite the higher acidity of thiocholine compared with **6** (pKa 7.2 vs 8.3)[46,47], its non-catalytic oxidation proceeded at the same rate as for **6** (Supplementary Fig. 12). This observation might indicate that a nucleophilic attack by thiolate might not be the sole rate-determining step in this reaction.

The catalytic oxidation of thiocholine also follows Michaelis–Menten-like kinetics (Fig. 8) but with multiple times smaller effective Michaelis constant than for **6**. This result is not totally unexpected. Based on the mechanism from Fig. 6, a high concentration of thiolate from thiocholine should lead to an increase in $k_1$ and a corresponding decrease in $k_2^*/k_1$ and $k_{-1}/k_1$ in Eq. (1). This change in rate constants should stabilize oscillations because it results in the

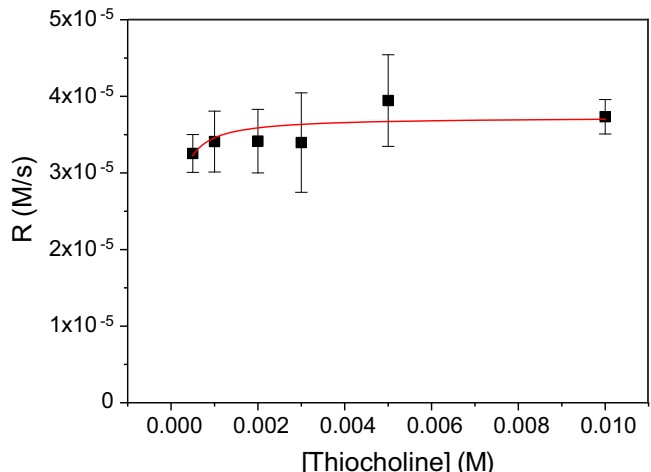

**Fig. 8 | Kinetic studies of the oxidation of thiocholine by 1 catalyzed by 2. [Se] = 2·[2].** All experiments were conducted in 1 M Tris pH 7.5, at 25 °C, [Se] = 0.2 mM, [**1**] = 7.5 mM, [thiocholine] = 0.5–10 mM. Data were fitted by equation $r = V_{max} \cdot S/(K_M + S)$: $V_{max} = 3.73 \pm 0.08 \cdot 10^{-5}\,M\,s^{-1}$, $K_M = 7.8 \pm 2.2 \cdot 10^{-5}\,M$. This fitting is approximate because the rate of catalytic oxidation showed no significant drop even at 0.5 mM concentration of thiocholine, and lowering the concentration even more is experimentally challenging. Nevertheless, the data show clearly that $K_M$ for thiocholine oxidation is significantly lower than for oxidation of **6**. Error bars represent standard deviations based on three independent experiments.

insensitivity of the oxidation rate to the concentration of thiols. Therefore, the conversion rate of thiols to disulfides decreases with the increase in the concentration of thiols, effectively increasing the nonlinearity of the negative feedback.

After implementing the Michaelis–Menten-like kinetics of oxidation with about ten times lower $k_2^*/k_1$ and $k_{-1}/k_1$ values for thiocholine than for **6**, we obtained sustained oscillations in the numerical model (Fig. 9, see Section 6 of the Supplementary Information for the details of the model). The model can generate sustained oscillation with a directly supplied active catalyst (the situation with catalyst **2**) (Fig. 9a) or with a catalyst generated in reaction with thiols from precursors (the situation with catalysts **3**–**5**) (Fig. 9b). The model produced sustained oscillations with experimentally used concentrations and experimentally determined rate constants for activation of pre-catalysts **3**–**5** (Fig. 9b and Supplementary Fig. 19). Importantly, the model also points towards the increased robustness of the oscillators with a negative feedback loop based on pre-catalysts **3**–**5**, compared to direct negative feedback with catalyst **2**.

The rate constants used to obtain sustained oscillations are not the same as in the independent kinetic measurements, but are still within 3x of the experimental values. The model also significantly (about 1.8 times) underestimates the oscillation periods. The limitations of the model are most likely related to many simplifications that were used in it. In particular, autocatalytic reactions with thiouronium salts are accompanied by the about stoichiometric release of protons and hence a small pH drop associated with limited buffer capacity. To account for the pH drop associated with autocatalysis and the partial (about 10%) deprotonation of **9**[29], we studied the kinetics of the oxidation of **6** at pH 7.5 instead of 7.7. However, the model does not account for periodic changes in pH. We briefly investigated the influence of the pH change from 7.7 to 7.5 on the rates of the catalytic and non-catalytic oxidation of **6** (Supplementary Table 1). Surprisingly, this small drop in pH had almost no effect on the rate of the non-catalytic reaction and appreciably increased the rate of catalytic reaction. Usually, an increase in pH always causes a drop in the rates of reactions involving thiols because of a decrease in the thiolate concentration, but aromatic selenols are strong enough acids (the pKa of selenophenol is 5.9)[48] to be almost fully deprotonated at the discussed pHs.

**a**

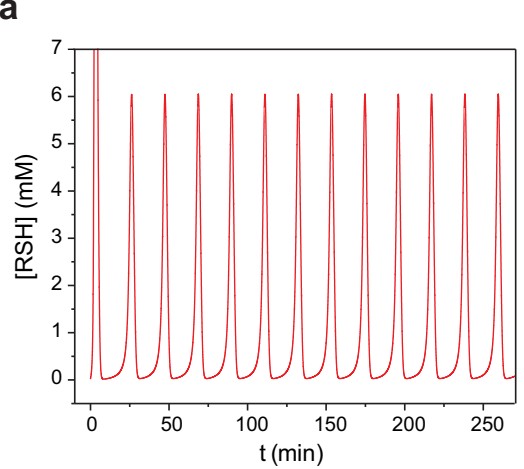

**b**

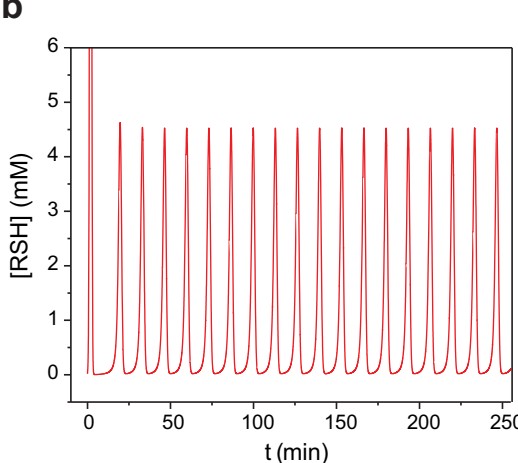

**Fig. 9 | Modeling of the oscillators. a** Simulations of the oscillations with catalyst **2**. The concentrations of reagents and the flow rate are as in the experiment shown in Fig. 4a. **b** Simulations of the oscillations with catalyst **5**. The concentrations of reagents and the flow rate are as in the experiment shown in Fig. 4d. Details of the model and the rate constants used can be found in Section 6 of Supplementary Information. RSH stands for the sum of concentrations of all three thiols in the system.

The rate increase might be related to the step of the protonation of *tert*-butylate, but precise elucidation of the reasons for this behavior requires a separate study. Nevertheless, even a small increase in the oxidation rate, associated with the pH drop during the autocatalytic step, is expected to stabilize oscillations because it increases the strength of the negative feedback.

Another important simplification hidden in Eq. (1) is the negligence of diselenide species in the catalytic cycle. Diselenide is considered to be in fast equilibrium with seleno-sulfide with equilibrium being shifted towards the latter. Although these assumptions are correct for the majority of conditions, they may not be accurate in situations when diselenide is constantly supplied and the concentration of thiols is similar to or lower than the concentration of diselenide. These conditions exist between spikes of oscillations in the system with catalyst **2**. Therefore, the model might be inaccurate in these situations. Nevertheless, we opted to keep the model at the current level of complexity to avoid over-parametrization, which will not aid in understanding this oscillating system.

## Discussion

Construction of systems and materials regulated by chemical reaction networks requires the availability of well-defined small sets of reactions – network motifs – that can be used modularly[3]. This work provides a negative feedback loop motif that complements the existing autocatalytic motifs[27,29,49,50] based on thiol chemistry. This negative feedback loop consists of two reactions: (i) substitution of the selenol from selenocarbonate by thiols. (ii) the selenol-catalyzed oxidation of thiols with organic hydroperoxides.

Based on the detailed kinetic studies, this negative feedback loop has several interesting features that should aid its future use. First, the rate of the selenol release from a selenocarbonate can be tuned by varying the substituent in the selenocarbonate. This tunability allows adjusting the strength of the negative feedback as needed for a desired task. Second, the selenol-catalyzed oxidation of thiols has Michaelis–Menten-like kinetics. The apparent Michaelis constant for this oxidation is in the millimolar range, which is compatible with typical concentrations in thiol-based reaction networks. In contrast to the first-order mass action kinetics, Michaelis–Menten kinetics is nonlinear in respect to the concentration of a substrate (thiols in this case). This nonlinearity might aid the emergence of instability-driven phenomena such as oscillations, waves, and patterns[32]. Thus, we experimentally confirmed the theoretical predictions[21,51] that the Michaelis–Menten kinetics of autocatalyst removal facilitates sustained oscillations. Notably, the dynamics of the system with catalyst **2** closely corresponds to the classic Higgins' model of enzyme-induced oscillations originally developed to explain oscillations in glycolysis[51,52].

Negative feedback loops are instrumental for constructing oscillatory, adaptive, and homeostatic chemical regulatory circuits and other emergent out-of-equilibrium phenomena[11,12,20]. We demonstrated the use of this negative feedback loop motif for obtaining emergent phenomena by combining it with the autocatalytic motif and obtaining sustained oscillations. This system is a unique example of the oscillator based on small organic molecules with a negative feedback loop instead of two direct inhibitors (Fig. 1). Combining the chemistry developed here with the regulation of hydrogel actuation[5,37,38,53–55], self-assembly[56,57], and liquid–liquid phase separation[58] will lead to materials with complex programmable responses[59,60].

## Methods

### Synthesis

For the complete synthetic procedures of compounds **2**–**5** we refer readers to Section 2 of the Supplementary information.

### Protocols for the batch kinetics experiments

Generally, the solution of **6** and diselenide catalyst **2** in Tris buffer pH 7.5 was mixed with the solution of **1**. The initial concentration of **6** was in the range of 0.5–10 mM; the concentration of diselenide catalyst **2** was in the range of 0.025–0.5 mM (corresponding to 0.05–1 mM of Se), and the initial **1** concentration was in the range of 3-39 mM. To follow the concentration of **6** in this reaction, we used the following experimental protocol: A freshly prepared solution of Ellman's reagent 2 mM in phosphate buffer solution (pH = 7.0; 200 mM) was used to fill UV–Vis cuvettes with 2 mL of solution in each cuvette. Next, 20 µL (or 10 µL) aliquots of the reaction mixture were taken every few seconds and mixed with Ellman's solution. The absorbance at 412 nm was measured by a UV–Vis spectrometer and converted to the total concentration of thiols.

To obtain the required plots, the absorbance data were analyzed as follows: First, the absorbance data were converted to concentrations of **6** in the reaction mixture using an extinction coefficient of $14{,}150 \ M^{-1} \ cm^{-1}$ and a dilution factor of 101 (20 µL aliquot vs 2020 µL of Ellman's solution plus aliquot) or 201 for 10 µL aliquot. Next, the concentration was plotted against time, and the initial linear regions

were identified and fitted with a linear function. The slopes of these lines represent the initial reaction rates. Measurements for each reaction rate point were repeated three times, and average values were used to make the required curves and to determine the reaction orders and the rate constants. Importantly, to analyze only the catalytic reaction, we subtracted the non-catalytic rate. Therefore, for each point of the se-catalyzed reaction rate, six measurements were made: three with catalyst **2** and three without it (in some cases background rates were calculated from the rate equation).

## Protocol for experiments in flow

A detailed description of the experimental setup used for flow experiments can be found in the _Section 3 of Supplementary Information and reference 27. Briefly, four syringes were filled with solutions. The first syringe contained compounds **8** (168 mM) and **1** (218 mM) in water. The second syringe contained one of the catalysts— **2** (0.6 mM), **3** (1.5 mM), **4** (3 mM), or **5** (6 mM) in water with 1% of DMF. The third syringe contained **9** (300 mM) in 3 M Tris buffer pH 7.5. The fourth syringe contained Ellman's reagent (16.6 mM) in the 2% solution of $KH_2PO_4$. The content of the first three syringes was supplied to a miniature (320 µL) continuous stirred tank reactor (CSTR) with required flows. The mixture that came out of CSTR was mixed with the content of the fourth syringe and analyzed using a flow cell connected to a UV–Vis spectrometer. The absorbance values were converted to thiol concentrations using the calibration curve (Supplementary Fig. 4).

## Data availability

The authors declare that all data supporting the findings of this study are available within the paper and its supplementary information files. Source data for Figs. 6 and 8 are provided in the Source Data file. All data are available from the corresponding author upon request. Source data are provided with this paper.

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

## Acknowledgements

This work was supported by the Israel Science Foundation (grants 2333/19 and 1562/23 to S.N.S.). This work was also supported by grant no. 2019-1075 to E.V.S. Priority 2030 is acknowledged for infrastructural support.

## Author contributions

S.N.S. supervised the research. S.N.S. and X.L. planned the project and designed experiments. X.L., P.F. and V.A.S. performed experiments. X.L., E.V.S. and S.N.S. analyzed data. S.N.S. performed numerical simulations. S.N.S. wrote the manuscript with help from B.S.S. and X.L.

## Competing interests

The authors declare no competing interests.
