## [Peer Review File · Nature Communications]

Selenium catalysis enables negative feedback organic oscillatorsREVIEWER COMMENTS

Reviewer #1 (Remarks to the Author):

The manuscript describes the discovery and analysis of a new type of negative feedback in oscillating chemical reaction networks, based on selenophenol-catalyzed oxidation of thiols. This is a very clever design, very well executed experimentally, and backed up by mechanistic investigations and kinetic modeling. One thing that may help to increase the impact of this work is to map (through modeling) the behavior of the system depending on parameters associated to the catalysis, and to compare this to experimental results. Besides that, this is great work. A few minor points remain:

In the SI: "163 of thiouronium salt 8" 163 what?

Try not to use "uL" for microliter, use the Greek mu instead.

In the abstract: "However, negative feedback loop motifs are unavailability" is not a proper sentence.

The spelling of Michaelis-Menten is incorrect on many occasions in the manuscript.

I'd refrain from naming the oscillator developed in one's own lab after oneself, that should be left to the community.

"but aromatic selenols are strong enough acids (the pKa of selenophenol is 5.9) to be fully deprotonated at the discussed pHs." Given the difference between the pH and pKa, a few percent may still be protonated. A slight rewording can solve this.

Reviewer #2 (Remarks to the Author):

This work by Semonov et al. developed a nonlinear negative feedback loop, RSH oxidation catalyzed by selenocarbonates, to replace the double negative feedbacks (slow and fast) for stable SH-species oscillatory rhythms. This is nice and original work of alternate network for designing organic oscillatory network. The oscillation period is about 20-30 min, shorter than Semonov- Whitesides oscillations (>60min, Nature 2016, 537, 656 - 660), for possible pulse – SH waves to drive active soft robot of continuous net displacement, as like BZ self-oscillating gel (Science 2006, 314, 798–801, Science Advance, 2020, 6, eaaz9125; PNAS, 2017, 114(33), 8704), rather than only homogeneous swelling or un-swelling. The work needs several revisions before publication in Nature Communications.

1. Equation-1 (line 246) loses a minus signal.

2. Please check the inconsistent sequence Figure-numbers between Supporting Information(SI) and the main text. Such as page 10-SI of the support information "Figure 2b main text" should be "Figure 3b main text", "(page 14-SI) are presented in Figure 5 in the main text" \diamond "are presented in Figure 6 in the main text", "(First paragraph of page 27-SI) The concentration of reagents in the incoming flow and flow/V values were taken as in the experiments (Figure 4 main text)" \diamond "The concentration of reagents in the incoming flow and flow/V values were taken as in the experiments (Figure 5 main text)"

3. This network in fact is the Higgins model (autocatalysis + enzyme negative loop), The original Higgins model and analysis before 40 years should be emphasized, i.e., J. Proc. Nat. Acad. Sci. U.S. 1964, 52, 989 and Ibanez, J. L.; Fair' em; and Velarde, M. G. Phys. Lett. 1976, 58A, 364.

4. It is pulse waves (Science 2006, 314, 798–801; Science Advance, 2020, 6, eaaz9125;

PNAS, 2017, 114(33), 8704; and Angew. Chem. Int. Ed. 2020, 59, 7106) rather than only front waves (Adv. Mater. 2021, e2106816), for continuous net locomotion. So this system is possible for short pulse waves if oscillatory period is decreased to less than 10 min, as like BZ oscillations. That is a good prospect for this work.

Reviewer #3 (Remarks to the Author):

The author consider an organic reaction network exhibiting oscillations. They expand the well-known thiol-based mechanism and introduce a pathway involving selenium compounds. The idea behind this is to allow the evolution of an autoinhibitory cycle that would lead to oscillations in the presence of the autocatalytic thiol chemistry. The presented reaction network reveals a new scenario that is unusual among oscillatory chemical reactions. In the classical construction of an oscillatory chemical system, autocatalysis interacts a simple negative feedback, and hence nonlinearity is only characteristic to the autocatalytic loop. In this experimental study, however, there is sufficient nonlinearity within the autoinhibitory loop. The manuscript is very well structured. Starting from a well-defined thiol chemistry, the authors logically and step-by-step introduce the necessary motifs (selenium compound in this case). The work involves a thorough kinetic study with proper data evaluation. The description, together with the supplementary material, and the results may be interesting to a wider audience. I recommend the manuscript for publication. I have the following comments that should be considered in the revised version.

- 1, The discussion following Fig. 1 should emphasize the difference in the network topology and focus more on the novelty of the negative feedback type introduced here.
- 2, The units should be checked throughout the text, they are missing at various places, especially in the caption of Fig. 6.
- 3, Fig. 8. should be rescaled so that the plot would occupy larger portion of the graph.

Referees' comments:

Referee #1 (Remarks to the Author):

The manuscript describes the discovery and analysis of a new type of negative feedback in oscillating chemical reaction networks, based on selenophenol-catalyzed oxidation of thiols. This is a very clever design, very well executed experimentally, and backed up by mechanistic investigations and kinetic modeling. One thing that may help to increase the impact of this work is to map (through modeling) the behavior of the system depending on parameters associated to the catalysis, and to compare this to experimental results. Besides that, this is great work.

Our Response: We thank the reviewer for his/her description of the manuscript. We indeed strived to not only construct a negative feedback loop for thiol chemistry but to also properly characterize its kinetic behavior. We thank the reviewer for the suggestion to study the behavior of the system depending on parameters associated with the catalysis. We, however, believe such a study to be beyond the scope of the current manuscript. Its proper execution will require the amount of work comparable in size to the current manuscript because (i) the problem is multidimensional; the catalytic step involves at least three independent rate constants, (ii) varying these parameters requires synthesis and implementation of new substances, (iii) system is nonlinear and highly sensitive to parameters in some regions requiring dense sampling with many experimental points with each point being three oscillatory experiments (for reproducibility). Nevertheless, we believe it is an interesting study that we will consider for future research.

In the SI: "163 of thiouronium salt 8" 163 what?

Our Response: We thank the referee for catching this point. Units, *mg*, were missing. Corrected.

Try not to use "uL" for microliter, use the Greek mu instead.

Our Response:

Corrected

In the abstract: “However, negative feedback loop motifs are unavailability” is not a proper sentence.

Our Response:

Corrected.

Abstract

Text removed: negative feedback loop motifs are unavailability for thiol chemistry.

Text added: negative feedback loop motifs are unavailable for thiol chemistry

The spelling of Michaelis-Menten is incorrect on many occasions in the manuscript

Our Response:

Corrected

I'd refrain from naming the oscillator developed in one's own lab after oneself, that should be left to the community.

Our Response:

We completely agree with the referee that reactions should be named by the community. We used the name Semenov-Whitesides oscillator because it was introduced by Epstein, who is a leading figure in the field of chemical oscillators, and coworkers in *J. Am. Chem. Soc.* **2023**, 145, 42, 23152–23159. Nevertheless, to avoid misunderstanding from the members of the community, we explicitly stated the source of the name in the manuscript.

P 3

Text removed: We have recently developed a series of thiol-based Semenov-Whitesides oscillators.²⁷⁻³⁰

Text added: We have recently developed a series of thiol-based oscillators,²⁷⁻²⁹ which are called Semenov-Whitesides oscillators by Epstein, Gao, and coworkers.³⁰

“but aromatic selenols are strong enough acids (the pKa of selenophenol is 5.9) to be fully deprotonated at the discussed pHs.” Given the difference between the pH and pKa, a few percent may still be protonated. A slight rewording can solve this.

Our Response:

We changed wording to “almost fully deprotonated”.

P 16

Text removed: but aromatic selenols are strong enough acids (the pKa of selenophenol is 5.9)⁴⁶ to be fully deprotonated at the discussed pHs..

Text added: but aromatic selenols are strong enough acids (the pKa of selenophenol is 5.9)⁴⁸ to be almost fully deprotonated at the discussed pHs..

Referee #2 (Remarks to the Author):

This work by Semonov et al. developed a nonlinear negative feedback loop, RSH oxidation catalyzed by selenocarbonates, to replace the double negative feedbacks (slow and fast) for stable SH-species oscillatory rhythms. This is nice and original work of alternate network for designing organic oscillatory network. The oscillation period is about 20-30 min, shorter than Semonov- Whitesides oscillations(>60min, Nature 2016, 537, 656 - 660), for possible pulse – SH waves to drive active soft robot of continuous net displacement, as like BZ self-oscillating gel (Science 2006, 314, 798–801, Science Advance, 2020, 6, eaaz9125; PNAS, 2017,114(33), 8704), rather than only homogeneous swelling or un-swelling.

Our Response:

We thank the reviewer for his/her remarks. Indeed, using this oscillating system to control hydrogel materials is one of the most promising continuations of this work, which we are currently exploring in the lab.

“Equation-1 (line 246) loses a minus signal.

Our Response:

Corrected

Please check the inconsistent sequence Figure-numbers between Supporting Information(SI) and the main text. Such as page 10-SI of the support information “Figure 2b main text” should be “Figure 3b main text”, “(page 14-SI)are presented in Figure 5 in the main text” □ “are presented in Figure 6 in the main text”, “(First paragraph of page 27-SI)The concentration of reagents in the incoming flow and flow/V values were taken as in the experiments (Figure 4 main text)” □“Theconcentration of reagents in the incoming flow and flow/V values were taken as in the experiments (Figure 5 main text)”

Our Response:

We thank the referee for noticing these inconsistencies. We have double-checked and revised the Figure numbers in Supporting Information to make it consistent with the main text.

This network in fact is the Higgins model (autocatalysis + enzyme negative loop), The original Higgins model and analysis before 40 years should be emphasized, i.e., J. Proc. Nat. Acad. Sci. U.S. 1964, 52, 989 and Ibanez, J. L.; Fair'em; and Velarde, M. G. Phys. Lett. 1976, 58A, 364.

Our Response:

We are very grateful to the referee for pointing out this scientifically very important point. We added a sentence that reflects this point and cited the mentioned manuscripts.

P 16

Text added: Notably, the dynamics of the system with catalyst 2 closely corresponds to the classic Higgins' model of enzyme-induced oscillations originally developed to explain oscillations in glycolysis.^{51, 52}

References

Text added: 51. Ibañez, J. L.; Fairén, V.; Velarde, M. G., Limit cycle and dissipative structures in a simple bimolecular nonequilibrium reactional scheme with enzyme. *Phys. Lett. A* 1976, 58, 364-366.

52. Higgins, J., A Chemical Mechanism for Oscillation of Glycolytic Intermediates in Yeast Cells. *Proc. Natl. Acad. Sci. U.S.A.* 1964, 51, 989-94.

It is pulse waves (Science 2006, 314, 798–801; Science Advance, 2020, 6, eaaz9125; PNAS, 2017,114(33), 8704; and Angew. Chem. Int. Ed. 2020, 59, 7106) rather than only front waves (Adv. Mater. 2021, e2106816), for continuous net locomotion. So this system is possible for short pulse waves if oscillatory period is decreased to less than 10 min, as like BZ oscillations. That is a good prospect for this work.

Our Response:

We are very grateful to the referee for pointing out the relevant literature. We cited it in the last sentence of the introduction. Moreover, in connection with the comment from the third referee, we introduced an additional sentence to the introduction.

P 3

Text added: These motifs would provide negative feedback that is more nonlinear than direct inhibition and enable oscillators with tunable frequency,^{11, 18} adaptive systems,¹² and various types of chemical waves and patterns that are instrumental for the synthesis of life-inspired materials.^{37, 38}

P 16

Text added: Combining the chemistry developed here with the regulation of hydrogel actuation,^{5, 37, 38, 53-55} self-assembly,^{56, 57} and liquid-liquid phase separation⁵⁸ will lead to materials with complex programmable responses.^{59, 60}

References

- Text added:* 37. Yashin, V. V.; Balazs, A. C., Pattern formation and shape changes in self-oscillating polymer gels. *Science* 2006, 314, 798-801.
38. Ren, L.; Wang, M.; Pan, C.; Gao, Q.; Liu, Y.; Epstein, I. R., Autonomous reciprocating migration of an active material. *Proc. Natl. Acad. Sci. U.S.A.* 2017, 114, 8704-8709.
54. Ren, L.; Wang, L.; Gao, Q.; Teng, R.; Xu, Z.; Wang, J.; Pan, C.; Epstein, I. R., Programmed Locomotion of an Active Gel Driven by Spiral Waves. *Angew. Chem. Int. Ed.* 2020, 59, 7106-7112.
55. Ren, L.; Yuan, L.; Gao, Q.; Teng, R.; Wang, J.; Epstein, I. R., Chemomechanical origin of directed locomotion driven by internal chemical signals. *Sci Adv* 2020, 6, eaaz9125.

Referee #3 (Remarks to the Author):

The author consider an organic reaction network exhibiting oscillations. They expand the well-known thiol-based mechanism and introduce a pathway involving selenium compounds. The idea behind this is to allow the evolution of an autoinhibitory cycle that would lead to oscillations in the presence of the autocatalytic thiol chemistry. The presented reaction network reveals a new scenario that is unusual among oscillatory chemical reactions. In the classical construction of an oscillatory chemical system, autocatalysis interacts a simple negative feedback, and hence nonlinearity is only characteristic to the autocatalytic loop. In this experimental study, however, there is sufficient nonlinearity within the autoinhibitory loop. The manuscript is very well structured. Starting from a well-defined thiol chemistry, the authors logically and step-by-step introduce the necessary motifs (selenium compound in this case). The work involves a thorough kinetic study with proper data evaluation. The description, together with the supplementary material, and the results may be interesting to a

wider audience. I recommend the manuscript for publication. I have the following comments that should be considered in the revised version.

Our Response: We thank the referee for precisely highlighting the key points of the manuscript. Indeed, the analysis of the kinetics of selenium-catalyzed oxidation of thiols can be interesting for readers outside of the fields of nonlinear chemical systems and systems chemistry.

The discussion following Fig. 1 should emphasize the difference in the network topology and focus more on the novelty of the negative feedback type introduced here.

Our Response:

We thank the referee for this valuable suggestion. We added text to the introduction section

P 3

Text added: Nevertheless, to expand the variety of dynamic behaviors and responses achievable with thiol-based reaction networks, it is essential to enrich the toolbox of available network motifs with negative feedback loops where thiols would trigger their elimination (Fig. 1b).¹² These motifs would provide negative feedback that is more nonlinear than direct inhibition and enable oscillators with tunable frequency,^{11, 18} adaptive systems,¹² and various types of chemical waves and patterns that are instrumental for the synthesis of life-inspired materials.^{37, 38}

The units should be checked throughout the text, they are missing at various places, especially in the caption of Fig. 6.

Our Response:

Corrected

Caption Figure 6

Text added: **b.** Dependence of the reaction rate on the concentration of **6** in experiments with different [Se] concentrations. [**1**] = 7.5 mM, [**6**] = 0.5 – 10 mM, [Se] are shown above each graph. Data for each set were fitted by equation $r = V_{max}S/(K_M+S)$: [Se] = 0.05 mM, $V_{max} = 1.5 \pm$

$0.1 \cdot 10^{-5} \text{ M}\cdot\text{s}^{-1}$, $K_M = 7.8 \pm 1.6 \cdot 10^{-4} \text{ M}$; $[\text{Se}] = 0.1 \text{ mM}$, $V_{max} = 2.8 \pm 0.2 \cdot 10^{-5} \text{ M}\cdot\text{s}^{-1}$, $K_M = 11 \pm 1.6 \cdot 10^{-4} \text{ M}$; $[\text{Se}] = 0.2 \text{ mM}$, $V_{max} = 4.5 \pm 0.2 \cdot 10^{-5} \text{ M}\cdot\text{s}^{-1}$, $K_M = 8.4 \pm 1.5 \cdot 10^{-4} \text{ M}$. (See Figure S11 for the full range of $[\text{Se}]$.) c. Dependence of the reaction rate on the concentration of **6** in experiments with different concentrations of **1**. $[\text{Se}] = 0.2 \text{ mM}$, $[\text{6}] = 0.5 - 10 \text{ mM}$, $[\text{1}]$ is shown above each graph. Data for each set were fitted by equation $r = V_{max} \cdot S / (K_M + S)$: $[\text{1}] = 3 \text{ mM}$, $V_{max} = 1.9 \pm 0.1 \cdot 10^{-5} \text{ M}\cdot\text{s}^{-1}$, $K_M = 3.8 \pm 0.8 \cdot 10^{-4} \text{ M}$; $[\text{1}] = 7.5 \text{ mM}$, $V_{max} = 4.5 \pm 0.2 \cdot 10^{-5} \text{ M}\cdot\text{s}^{-1}$, $K_M = 8.4 \pm 1.5 \cdot 10^{-4} \text{ M}$; $[\text{1}] = 12 \text{ mM}$, $V_{max} = 6.9 \pm 0.3 \cdot 10^{-5} \text{ M}\cdot\text{s}^{-1}$, $K_M = 14.5 \pm 1.0 \cdot 10^{-4} \text{ M}$.

Caption Figure 8

Text added: Kinetic studies of the oxidation of thiocholine by **1** catalyzed by **2**. $[\text{Se}] = 2 \cdot [\text{2}]$. All experiments were conducted in 1 M Tris pH 7.5, at 25 °C, $[\text{Se}] = 0.2 \text{ mM}$, $[\text{1}] = 7.5 \text{ mM}$, $[\text{thiocholine}] = 0.5 - 10 \text{ mM}$. Data were fitted by equation $r = V_{max} \cdot S / (K_M + S)$: $V_{max} = 3.73 \pm 0.08 \cdot 10^{-5} \text{ M}\cdot\text{s}^{-1}$, $K_M = 7.8 \pm 2.2 \cdot 10^{-5} \text{ M}$.

Fig. 8. should be rescaled so that the plot would occupy larger portion of the graph.

Our Response:

Indeed, this will be a standard way of formatting this plot. However, the figure has to illustrate the difference in the behavior of cysteamine and thiocholine. By showing the zero-point, we simplify the comparison with analogous plots for cysteamine from Figure 6 and better visualize the weak dependence of the oxidation rate from the concentration of thiocholine. Therefore, we opted to not rescale this plot.

REVIEWERS' COMMENTS

Reviewer #2 (Remarks to the Author):

The authors have addressed my concerns. I agree to publish the manuscript in Nature Communications.

Reviewer #3 (Remarks to the Author):

The author have answered my questions. With the included modifications I recommend the revised manuscript for publication.